# Deep Reinforcement Learning for the Control of Robotic Manipulation: A Focussed Mini-Review

**Rongrong Liu** [1],*[iD]**, Florent Nageotte** [1]**, Philippe Zanne** [1]**, Michel de Mathelin** [1] **and Birgitta Dresp-Langley** [2],*[iD]

1   ICube Lab, Robotics Department, Strasbourg University, UMR 7357 CNRS, 67085 Strasbourg, France; Nageotte@unistra.fr (F.N.); philippe.zanne@unistra.fr (P.Z.); demathelin@unistra.fr (M.d.M.)
2   ICube Lab, UMR 7357, Centre National de la Recherche Scientifique CNRS, 67085 Strasbourg, France
*   Correspondence: rongrong.liu@unistra.fr (R.L.); birgitta.dresp@unistra.fr (B.D.-L.)

**Abstract:** Deep learning has provided new ways of manipulating, processing and analyzing data. It sometimes may achieve results comparable to, or surpassing human expert performance, and has become a source of inspiration in the era of artificial intelligence. Another subfield of machine learning named reinforcement learning, tries to find an optimal behavior strategy through interactions with the environment. Combining deep learning and reinforcement learning permits resolving critical issues relative to the dimensionality and scalability of data in tasks with sparse reward signals, such as robotic manipulation and control tasks, that neither method permits resolving when applied on its own. In this paper, we present recent significant progress of deep reinforcement learning algorithms, which try to tackle the problems for the application in the domain of robotic manipulation control, such as sample efficiency and generalization. Despite these continuous improvements, currently, the challenges of learning robust and versatile manipulation skills for robots with deep reinforcement learning are still far from being resolved for real-world applications.

**Keywords:** deep learning; artificial intelligence; machine learning; reinforcement learning; deep reinforcement learning; robotic manipulation control; sample efficiency; generalization





## 1. Introduction

Robots were originally designed to assist or replace humans by performing repetitive and/or dangerous tasks which humans usually prefer not to do, or are unable to do because of physical limitations imposed by extreme environments. Those include the limited accessibility of narrow, long pipes underground, anatomical locations in specific minimally invasive surgery procedure, objects at the bottom of the sea, for example. With the continuous developments in mechanics, sensing technology [1], intelligent control and other modern technologies, robots have improved autonomy capabilities and are more dexterous. Nowadays, commercial and industrial robots are in widespread use with lower long-term cost and greater accuracy and reliability, in the 2 fields like manufacturing, assembly, packing, transport, surgery, earth and space exploration, etc.

There are different types of robots available, which can be grouped into several categories depending on their movement, Degrees of Freedom (DoF), and function. Articulated robots, are among the most common robots used today. They look like a human arm and that is why they are also called robotic arm or manipulator arm [2]. In some contexts, a robotic arm may also refer to a part of a more complex robot. A robotic arm can be described as a chain of links that are moved by joints which are actuated by motors. We will start from a brief explanation of these mechanical components of a typical robotic manipulator [3,4]. Figure 1 shows the schematic diagram of a simple two-joint robotic arm mounted on a stationary base on the floor .

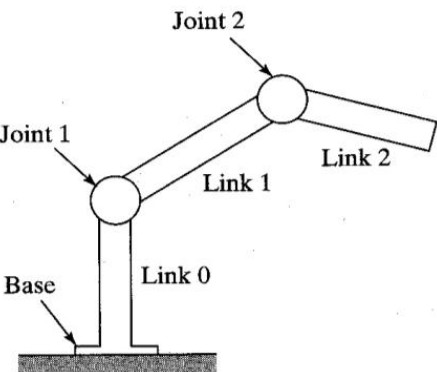

**Figure 1.** Simplified schematic diagram of mechanical components of a two-joint robotic arm.

- Joints are similar to joints in the human body, which provide relative motion between two parts of the body. In robotic field, each joint is a different axis and provides an additional DoF of controlled relative motion between adjacent links, as shown in Figure 1. In nearly all cases, the number of degrees of freedom is equal to the number of joints [5].
- An end-effector, as the name suggests, is an accessory device or tool which is attached at the end of the chain, and actually accomplishes the intended task. The simplest end-effector is the gripper, which is capable of opening and closing for gripping objects, but it can also be designed as a screw driver, a brush, a water jet, a thruster, or any mechanical device, according to different applications. An end-effector can also be called a robotic hand.
- Links are the rigid or nearly rigid components that connect either the base, joints or end effector, and bear the load of the chain.
- An actuator is a device that converts electrical, hydraulic, or pneumatic energy into robot motion.

Currently, the control sequence of a robotic manipulator is mainly achieved by solving inverse kinematic equations to move or position the end effector with respect to the fixed frame of reference [6,7]. The information is stored in memory by a programmable logic controller for fixed robotic tasks [8]. Robots can be controlled in open-loop or with an exteroceptive feedback. The open-loop control does not have external sensors or environment sensing capability, but heavily relies on highly structured environments that are very sensitively calibrated. If a component is slightly shifted, the control system may have to stop and to be recalibrated. Under this strategy, the robot arm performs by following a series of positions in memory, and moving to them at various times in their programming sequence. In some more advanced robotic systems, exteroceptive feedback control is employed, through the use of monitoring sensors, force sensors, even vision or depth sensors, that continually monitor the robot's axes or end-effector, and associated components for position and velocity. The feedback is then compared to information stored to update the actuator command so as to achieve the desired robot behavior. Either auxiliary computers or embedded microprocessors are needed to perform interface with these associated sensors and the required computational functions. These two traditional control scenarios are both heavily dependent on hardware-based solutions. For example, conveyor belts and shaker tables, are commonly used in order to physically constrain the situation.

With the advancements in modern technologies in artificial intelligence, such as deep learning, and recent developments in robotics and mechanics, both the research and industrial communities have been seeking more software based control solutions using low-cost sensors, which has less requirements for the operating environment and calibration. The key is to make minimal but effective hardware choices and focus on robust algorithms and software. Instead of hard-coding directions to coordinate all the joints, the control policy could be obtained by learning and then be updated accordingly. Deep

Reinforcement Learning (DRL) is among the most promising algorithms for this purpose because no predefined training dataset is required, which ideally suits robotic manipulation and control tasks, as illustrated in Table 1. A reinforcement learning approach might use input from a robotic arm experiment, with different sequences of movements, or input from simulation models. Either type of dynamically generated experiential data can be collected, and used to train a Deep Neural Network (DNN) by iteratively updating specific policy parameters of a control policy network.

**Table 1.** Comparison between traditional control and DRL based control expectation.

|  | **Traditional Control** | **DRL Based Control Expectation** |
| --- | --- | --- |
| control solution | hardware based | software based |
| monitoring sensor | expensive | low-cost |
| environment requirement | structured | unstructured situations |
| hardware calibration | sensitive to calibration | tolerate to calibration |
| control algorithm | hand coding required | data driven |

This review paper tries to provide a brief and self-contained review of DRL in the research of robotic manipulation control. We will start with a brief introduction of deep learning, reinforcement learning, and DRL in the second section. The recent progress of robotic manipulation control with the DRL based methods will be then discussed in the third section. What need to mention here is that, we can not cover all the brilliant algorithms in detail in a short paper. For the algorithms mentioned here, one still need to refer to those original papers for the detailed information. Finally, we follow the discussion and present other real-world challenges of utilizing DRL in robotic manipulation control in the forth section, with a conclusion of our work in the last section.

## 2. Deep Reinforcement Learning

In this part, we will start from deep learning and reinforcement learning, to better illustrate their combination version, DRL.

### 2.1. Deep Learning

Deep learning is quite popular in the family of machine learning, with its outstanding performance in a variety of domains, not only in classical computer vision tasks, but also in many other practical applications; to just name a few, natural language processing, social network filtering, machine translation, bioinformatics, material inspection and board games, where these deep-learning based methods have produced results comparable to, and in some cases surpassing human expert performance. Deep learning has changed the way we process, analyze and manipulate data.

The adjective "deep" in deep learning comes from the use of multiple layers in the network. Figure 2 demonstrates a simple deep learning architecture with basic fully connected strategy. A general comparison is conducted in Table 2 between deep learning and traditional machine learning. With deep learning, the raw data like images are directly fed into a deep neural network multiple layers that progressively extract higher-level features, while with traditional machine learning, the relevant features of input data are manually extracted by experts. Besides, deep learning often requires a large amount of data to reach optimal results, thus it is also computationally intensive.

**Table 2.** Comparison between traditional machine learning and deep learning.

|  | **Traditional Machine Learning** | **Deep Learning** |
|---|---|---|
| dataset requirement | performs well with small dataset | requires large dataset |
| accuracy | accuracy plateaus | excellent performance potential |
| feature extraction | selected manually | learned automatically |
| algorithm structure | simple model | multi-layer model |
| model training time | quick to train a model | computationally intensive |
| hardware requirement | works with not powerful hardware | high-performance computer |

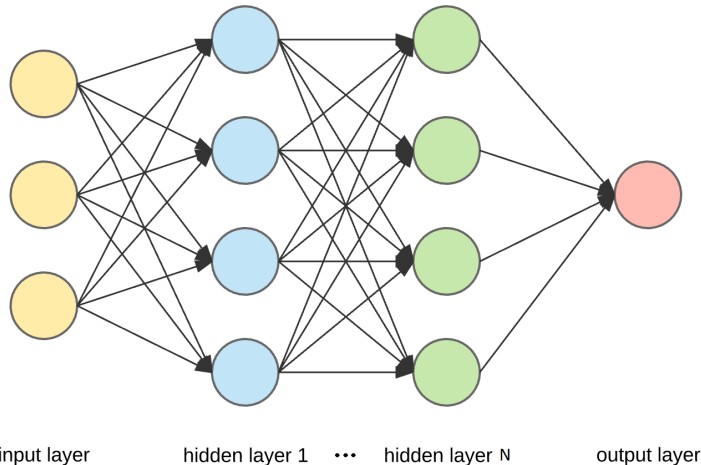

input layer    hidden layer N    ⋯    hidden layer N    output layer

**Figure 2.** A simple deep learning architecture.

Deep models can be interpreted as artificial neural networks with deep structures. The idea of artificial neural networks is not something new, which can date back to 1940s [9]. In the following years, the research community witnessed many important milestones with perceptrons [10], backpropagation algorithm [11,12], Rectified Linear Unit, or ReLU [13], Max-pooling [14], dropout [15], batch normalization [16], etc. It is all these continuous algorithmic improvements, together with the emergence of large-scale training data and the fast development of high-performance parallel computing systems, such as Graphics Processing Units (GPUs) that allow deep learning to prosper nowadays [17].

The first great success for deep learning is based on a convolutional neural network for classification in 2012 [18]. It applies hundreds of thousands data-label pairs iteratively to train the parameters with loss computation and backpropagation. Although this technique has been improved continuously and rapidly since it took off, and is now one of the most popular deep learning structures, it is not quite suitable for robotic manipulation control, as it is too time-consuming to obtain large number of images of joints angles with labeled data to train the model. Indeed, there are some researches using convolutional neural network to learn the motor torques needed to control the robot with raw RGB video images [19]. However, a more promising and interesting idea is using DRL, as we will discuss hereafter.

*2.2. Reinforcement Learning*

Reinforcement learning [20] is a subfield of machine learning, concerned with how to find an optimal behavior strategy to maximize the outcome though trial and error dynamically and autonomously, which is quite similar with the intelligence of human and animals, as the general definition of intelligence is the ability to perceive or infer information, and to retain it as knowledge to be applied towards adaptive behaviors in the environment. This autonomous self-teaching methodology is actively studied in many

domains, like game theory, control theory, operations research, information theory, system optimization, recommendation system and statistics [21].

Figure 3 illustrates the universal model of reinforcement learning, which is biologically plausible, as it is inspired by learning through punishment or reward due to state changes in the environment, which are either favorable (reinforcing) to certain behaviors/actions, or unfavourable (suppressing). Natural reinforcement learning is driven by the evolutionary pressure of optimal behavioral adaptation to environmental constraints.

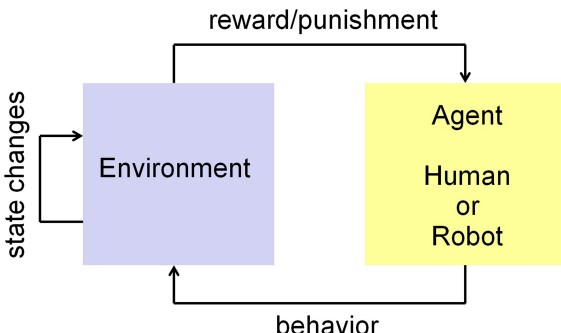

**Figure 3.** Universal model of reinforcement learning.

When an agent is in a state, it chooses an action according to its current policy and then it receives a reward from the environment for executing that action. By learning from this reward, it transitions to a new state, chooses a new action and then iterates through this process. In order to make it even easier to understand, one can compare reinforcement learning to the structure of how we play a video game, in which the character, namely the agent, engages in a series of trials, or actions, to obtain the highest score, which is reward.

Reinforcement learning is different from supervised learning, where a training set of labeled examples is available. In interactive problems like robot control domain using reinforcement learning, it is often impractical to obtain examples of desired behavior that are both correct and representative of all the situations in which the agent has to act. Instead of labels, we get rewards which in general are weaker signals. Reinforcement learning is not a kind of unsupervised learning, which is typically about finding structure hidden in collections of unlabeled data. In reinforcement learning, the agent has to learn to behave in the environment based only on those sparse and time-delayed rewards, instead of trying to find hidden structure. Therefore, reinforcement learning can be considered as a third machine learning paradigm, alongside supervised learning and unsupervised learning and perhaps other future paradigms as well [22].

*2.3. Deep Reinforcement Learning*

As the name suggests, DRL emerges from reinforcement learning and deep learning, and can be regarded as the bridge between conventional machine learning and true artificial intelligence, as illustrated in Figure 4. It combines both the technique of giving rewards based on actions from reinforcement learning, and the idea of using a neural network for learning feature representations from deep learning. Traditional reinforcement learning is limited to domains with simple state representations, while DRL makes it possible for agents to make decisions from high-dimensional and unstructured input data [23] using neural networks to represent policies. In the past few years, research in DRL has been highly active with a significant amount of progress, along with the rising interest in deep learning.

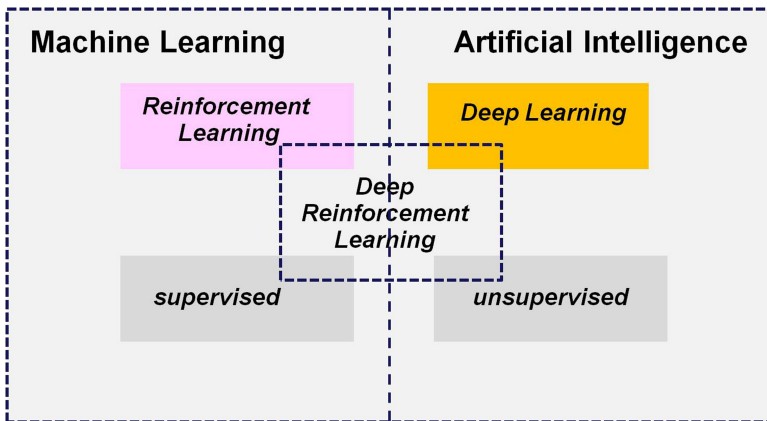

**Figure 4.** Deep reinforcement learning.

FDRL has gained a lot of attraction, especially due to its well-known achievement in games. Beginning around 2013, DeepMind showed impressive learning results in Atari video games at human level proficiency with no hand coded features using unprocessed pixels for input [24,25], which can be regarded as the creation of this subfield. Another milestone was in 2016 when AlphaGo [26] first beat a human professional player of Go, which is a game from ancient China. This computer program was improved by AlphaZero [27] in 2017, together with its efficiency also in chess and shogi. In 2019, Pluribus [28] showed its success over top human professionals in multiplayer poker, and OpenAI Five [29] beat the previous world champions in a Dota 2 demonstration match.

Apart from the field of games, it has large potential in other domains, including but not limited to, robotics [30], natural language processing [31], computer vision [32], transportation [33], finance [34] and healthcare [35]. Many exciting breakthroughs of this research have been published by both of giant companies, which include Google Brain, DeepMind and Facebook, and top academic labs such as in Berkeley, Stanford and Carnegie Mellon University, together with some independent non-profit research organizations like openAI and some other industrially focused companies.

The most commonly used DRL algorithms can be categorized in value-based methods, policy gradient methods and model-based methods. The value-based methods construct a value function for defining a policy, which is based on the Q-learning algorithm [36] using the Bellman equation [37] and its variant, the fitted Q-learning [38,39]. The Deep Q-Network (DQN) algorithm used with great success in [25] is the representative of this class, followed by various extensions, such as double DQN [40], Distributional DQN [41,42], etc. A combination of these improvements has been studied in [43] with a state-of-the-art performance on the Atari 2600 benchmark, both in terms of data efficiency and final performance.

However, the DQN-based approaches are limited to problems with discrete and low-dimensional action spaces, and deterministic policies, while policy gradient methods are able to work with continuous action spaces and can also represent stochastic policies. Thanks to variants of stochastic gradient ascent with respect to the policy parameters, policy gradient methods are developed to find a neural network parameterized policy to maximize the expected cumulative reward [44]. Like other policy-based methods, policy gradient methods typically require an estimate of a value function for the current policy and a sample efficient approach is to use an actor-critic architecture that can work with off-policy data. The Deep Deterministic Policy Gradient (DDPG) algorithm [45,46] is a representation of this type of methods. There are also some researchers working on combining policy gradient methods with Q-learning [47].

Both value-based and policy-based methods do not make use of any model of the environment and are also called model-free methods, which limits their sample efficiency. On the contrary, in the model-based methods, a model of the environment is either explicitly given or learned from experience by the function approximators [48,49] in conjunction

with a planning algorithm. In order to obtain advantages from both sides, there are many researches available integrating model-free and model-based elements [50–52], which are among the key areas for the future development of DRL algorithms [53].

### 3. Deep Reinforcement Learning in Robotic Manipulation Control

In this section, the recent progress of DRL in the domain of robotic manipulation control will be discussed. Two of the most important challenges here concern sample efficiency and generalization. The goal of DRL in the context of robotic manipulation control is to train a deep policy neural network, like in Figure 2, to detect the optimal sequence of commands for accomplishing the task. As illustrated in Figure 5, the input is the current state, which can include the angles of joints of the manipulator, position of the end effector, and their derivative information, like velocity and acceleration. Moreover, the current pose of target objects can also be counted in the current state, together with the state of corresponding sensors if there are some equipped in the environment. The output of this policy network is an action indicating control commands to be implemented to each actuator, such as torques or velocity commands. When the robotic manipulator accomplishes a task, a positive reward will be generated. With these delayed and weak signals, the algorithm is expected to find out the most successful control strategy for the robotic manipulation.

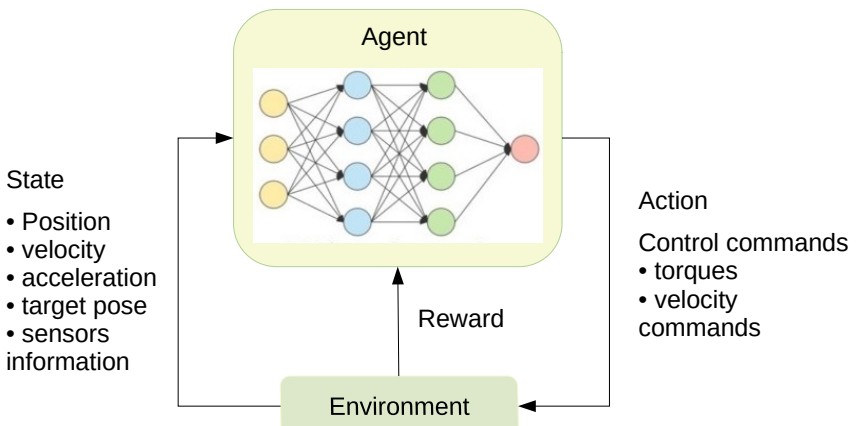

**Figure 5.** A schematic diagram of robotic manipulation control using DRL.

### 3.1. Sample Efficiency

As we know, in supervised deep learning, a training set of input-output pairs are fed to neutral networks to construct an approximation that maps an input to an output [54,55]. This learned target function can then be used for labeling new examples when a test set is given. The study of sample efficiency for supervised deep learning tries to answer the question of how large a training set is required in order to learn a good approximation to the target concept [56]. Accordingly, for DRL in robotic manipulation control, the study of sample efficiency discusses how much data need to be collected in order to build an optimal policy to accomplish the designed task.

The sample efficiency for DRL is considerably more challenging than that for supervised learning for various reasons [57]. First, the agent can not receive a training set provided by the environment unilaterally, but information which is determined by both the actions it takes and dynamics of the environment. Second, although the agent desires to maximize the long-term reward, the agent can only observe the immediate reward. Additionally, there is no clear boundary between training and test phases. The time the agent spends trying to improve the policy often comes at the expense of utilizing this policy, which is often referred to as the exploration–exploitation trade-off [53].

Since gathering experiences by interacting with the environment for robots is relatively expensive, a number of approaches have been proposed in the literature to address sample efficient learning. For example, in [58], which is the first demonstration of using DRL on a robot, trajectory optimization techniques and policy search methods with neural networks were applied to achieve reasonable sample efficient learning. A range of dynamic manipulation behaviors were learned, such as stacking large Lego blocks at a range of locations, threading wooden rings onto a tight-fitting peg, screwing caps onto different bottles, assembling a toy airplane by inserting the wheels into a slot. Videos and other supplementary materials can be found online [59]. Millimeter level precision can be achieved with dozens of examples by this algorithm, but the knowledge of the explicit state of the world at training time is required to enable sample efficiency.

In [60], a novel technique called Hindsight Experience Replay (HER) was proposed. Each episode was replayed but with a different goal than the one the agent was trying to achieve. With this clever strategy for augmenting samples, the policy for the pick-and-place task, which was learned using only sparse binary rewards, performed well on the physical robot without any finetuning. The video presenting their experiments is available online [61]. But this algorithm relies on special value function approximators, which might not be trivially applicable to every problem. Besides, this technique can not be extended well with the use of reward shaping.

Some other researchers try to achieve sample efficient learning through demonstrations in imitation learning [62,63], where a mentor provides demonstrations to replace the random neural network initialization. [64] was an extension of DDPG algorithm [46] for tasks with sparse rewards, where both demonstrations and actual interactions were used to fill a replay buffer. Experiments of four simulation tasks and a real robot clip insertion problem were conducted. A video demonstrating the performance can be viewed online [65]. However, the object location and the explicit states of joints, such as position and velocity, must be provided to move from simulation to real-world, which limits its application to high-dimensional data.

Based on the work of generative adversarial imitation learning in [66,67] used Generative Adversarial Networks (GANs) [68] to generate an additional training data to solve sample complexity problem, by proposing a multi-modal imitation learning framework that was able to handle unstructured demonstrations of different skills. The performance of this framework was evaluated in simulation for several tasks, such as reacher and gripper-pusher. The video of simulated experiments is available online [69]. Like most GANs techniques, it is quite difficult to train and many samples are required.

### 3.2. Generalization

Generalization, refers to the capacity to use previous knowledge from a source environment to achieve a good performance in a target environment. It is widely seen as a step necessary to produce artificial intelligence that behaves similarly to humans. Generalization may improve the characteristics of learning the target task by increasing the starting reward on the target domain, the rate of learning for the target task, and the maximum reward achievable [70].

In [71], Google proposed a method to learn hand–eye coordination for robot grasp task. In the experiment phrase, they collected about 800,000 grasp attempts over two months from multiple robots operating simultaneously, and then used these data to train a controller that work across robots. These identical uncalibrated robots had differences in camera placement, gripper wear or tear. Besides, a second robotic platform with eight robots collected a dataset consisting of over 900,000 grasp attempts, which was used to test transfer between robots. The results of transfer experiment illustrated that data from different robots can be combined to learn more reliable and effective grasping. One can refer to the video online [72] for supplementary results. In contrast to many prior methods, there is no simulation data or explicit representation, but an end-to-end training directly from image pixels to gripper motion in task space by learning just from this high-dimensional

representation. Despite of its attractive success, this method still can not obtain satisfactory accuracy for real application, let alone it is very hardware and data intensive.

To develop generalization capacities, some researchers turn to meta learning [73], which is also known as learning to learn. The goal of meta learning is to train a model on a variety of learning tasks, such that it can solve new learning tasks using only a small number of training samples [74]. In [75], meta learning was combined with aforementioned imitation learning in order to learn to perform tasks quickly and efficiently in complex unstructured environments. The approach was evaluated on planar reaching and pushing tasks in simulation, and visual placing tasks on a real robot, where the goal is to learn to place a new object into a new container from a single demonstration. The video results are available online [76]. The proposed meta-imitation learning method allows a robot to acquire new skills from just a single visual demonstration, but the accuracy needs to be further improved.

There are also many other researches available to tackle other challenges in this domain. For example, no matter whether a control policy is learned directly for a specific task, or transferred from previous tasks, another important but understudied question is how well will the policy performs, namely policy evaluation problem. A behavior policy search algorithm was proposed in [77] for more efficiently estimating the performance of learned policies.

## 4. Discussion

Although algorithms of robotic manipulation control using DRL have been emerging in large numbers in the past few years, some even with demonstration videos showing how an experimental robotic manipulator accomplishes a task with the policy learned, as we have illustrated above, the challenges of learning robust and versatile manipulation skills for robots with DRL are still far from being resolved satisfactorily for real-world application.

Currently, robotic manipulation control with DRL may be suited to fault tolerant tasks, like picking up and placing objects, where a disaster will not be caused if the operation fails occasionally. It is quite attractive in situations, where there is enough variation that the explicit modeling algorithm does not work. Potential applications can be found in warehouse automation to replace human pickers for objects of different size and shape, clothes and textiles manufacturing, where cloth is difficult to manipulate by nature, and food preparation industry, where, for example, every chicken nugget looks different, and it is not going to matter terribly if a single chicken nugget is destroyed.

However, even in this kind of applications, DRL-based methods are not widely used in real-world robotic manipulation. The reasons are multiple, including the two concerns we have discussed in the previous section, sample efficiency and generation, where more progress is still required, as both gathering experiences by interacting with the environment and collecting expert demonstrations for imitation learning are expensive procedures, especially in situations where robots are heavy, rigid and brittle, and it will cost too much if the robot is damaged in exploration. Another very important issue is safety guarantee. Not like simulation tasks, we need to be very careful that learning algorithms are safe, reliable and predictable in real scenarios, especially if we move to other applications that require safe and correct behaviors with high confidence, such as surgery or household robots taking care of the elder or the disabled. There are also other challenges including but not limited to the algorithm explainability, the learning speed, high-performance computational equipment requirements.

## 5. Conclusions

The scalability of DRL, discussed and illustrated here, is well-suited for high-dimensional data problems in a variety of domains. In this paper, we have presented a brief review of the potential of DRL for policy detection in robotic manipulation control and discussed the current research and development status of real-world applications. Through a joint development of deep learning and reinforcement learning, with inspiration from other

machine learning methods like imitation learning, GANs, or meta learning, new algorithmic solutions can emerge, and are still needed, to meet challenges in robotic manipulation control for practical applications.

**Author Contributions:** Conceptualization, R.L. and B.D.-L.; methodology, R.L. and B.D.-L.; investigation, R.L., F.N., P.Z. and B.D.-L.; resources, R.L., F.N., P.Z. and B.D.-L.; writing—original draft preparation, R.L.; writing—review and editing, F.N., P.Z. and B.D.-L.; supervision, B.D.-L.; project administration, B.D.-L. and M.d.M.; funding acquisition, B.D.-L. and M.d.M. All authors have read and agreed to the published version of the manuscript.

**Funding:** This research work is part of a project funded by the University of Strasbourg's Initiative D'EXellence (IDEX).

**Institutional Review Board Statement:** Not applicable.

**Informed Consent Statement:** Not applicable.

**Acknowledgments:** The support of the CNRS is gratefully acknowledged.

**Conflicts of Interest:** The authors declare no conflict of interest.

## Abbreviations

The following abbreviations are used in this manuscript:

| | |
|---|---|
| DoF | Degrees of Freedom |
| DRL | Deep Reinforcement Learning |
| DNN | Deep Neural Network |
| DQN | Deep Q-Network |
| DDPG | Deep Deterministic Policy Gradient |
| HER | Hindsight Experience Replay |
| GANs | Generative Adversarial Networks |

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
