# Peer review of "Deep Reinforcement Learning for the Control of Robotic Manipulation: A Focussed Mini-Review"

_robotics, doi:10.3390/robotics10010022_

Round 1

Reviewer 1 Report

This paper presents recent significant progress of deep reinforcement learning algorithms, which try to tackle the problems for the application in the domain of robotic manipulation control, such as sample efficiency and generalization. The authors draw the conclusion that the challenges of learning robust and versatile manipulation skills for robots with deep reinforcement learning are still far from being resolved for real-world applications.

The paper is well organized and written, the description of theories and techniques is clear and correct. Though the high quality of the work, there are still some aspects that need to be improved.

  1. the references are not new enough that reflect the latest progress of DLR in robotics
  2. Instead of explanations of the fundamental concepts and principles, the authors should give more insights of the topic

Author Response

#1: Reviewer’s comments:

The paper is well organized and written, the description of theories and techniques is clear and correct. Though the high quality of the work, there are still some aspects that need to be improved.

  1. The references are not new enough that reflect the latest progress of DLR in robotics.

Authors’ response: In the manuscript section 3, most recent progress of deep reinforcement learning in the domain of robotic manipulation control has been discussed, concerning two important challenges, sample efficiency and generalization. we have discussed some attractive new work done, including pre-printed papers from Arxiv, some of which are still not available in other conferences or journals, like [62] and [70].

  1. Instead of explanations of the fundamental concepts and principles, the authors should give more insights of the topic.

Authors’ response: We hope to offer a self-contained mini review that even suits readers who are not in this domain but interested, or students/beginners. That is why we have explained the fundamental concepts and principles for readers of different backgrounds

Reviewer 2 Report

The content of the article is consistent with the scientific area of the Journal of Robotics. The subject raised by the authors is current and so far rarely noticed by other authors publishing in this area.
The issue described may in the future contribute to improving the efficiency of the automation, control algorithms for manipulators and mobile robot, sensors for robot state sensing, part handling, navigation andsensors machine vision.
However, it should be noted that the authors should make some corrections in the article.
For a better clarification, please edit your paper as follows:

  1. Extend the text of manuscript (example introduction or conclusion) to concrete results in the world and in Europe, - Improve the quality of the paper by presenting the results of publications of researchers and experts that are registered in the world databases (wos). They are specifically these:
    Navigation control and stability investigation of a mobile robot based on a hexacopter equipped with an integrated manipulator
  2. figures 8 and 9 should be contrasting and readable,
  3. conclusions and future work should be extended to contain practical applications based on research described in this paper - expand references,
  4. highlight the course of dependencies/relations in figure No. 2 and 4,
  5. the paper should be read by a native english speaker.
    After consideration of these minor comments, the article is properly prepared (in the reviewer opinion) for publication in the Journal of Robotics.

Author Response

#2: Reviewer’s comments:

For a better clarification, please edit your paper as follows:

1. Extend the text of manuscript (example introduction or conclusion) to concrete results in the world and in Europe, - Improve the quality of the paper by presenting the results of publications of researchers and experts that are registered in the world databases (wos). They are specifically these:
Navigation control and stability investigation of a mobile robot based on a hexacopter equipped with an integrated manipulator

Authors’ response: We sincerely appreciate the positive comments. In this manuscript, we try to offer a focused mini-review of deep reinforcement learning for robotic manipulation control. Based on the introduction of the related techniques, we focus on the continuous efforts researchers have conducted to deal with two major challenges, namely sample efficiency and generalization. The researchers involved here have developed their algorithms with their own experimental platform or with the help of some simulation software, which are not evaluated with a universally known dataset. As the reviewer suggested, we have added the reference entitled “Navigation control and stability investigation of a mobile robot based on a hexacopter equipped with an integrated manipulator” in the introduction section.

2. figures 8 and 9 should be contrasting and readable,

Authors’ response:After discussion, we finally realized Figures 8 and 9 are not necessary for understanding our review paper. To avoid copyright issues, we have removed the Figures 8 and 9 in the revised version.

3. conclusions and future work should be extended to contain practical applications based on research described in this paper - expand references.

Authors’ response: As this technique of robotic manipulation control with deep reinforcement learning is still in the “lab stage”, and not qualified for practical applications, we can only offer some possible but promising application domains, now discussed with a little more detail in the manuscript Section 4.

4. highlight the course of dependencies/relations in figure No. 2 and 4.

Authors’ response: Thanks for your advice. Figure 2 shows a simple deep learning architecture. Figure 4 (Figure 5 in the revised version) is a schematic diagram of deep reinforcement learning for robotic manipulation control, where the agent is represented by a deep neural network. We have rephrased the sentence as “...train a deep policy neural network, like in Figure 2, to detect...”. Please kindly refer to lines 196-198 in the first paragraph in Section 3, on Page 6.

5. the paper should be read by a native English speaker.
After consideration of these minor comments, the article is properly prepared (in the reviewer opinion) for publication in the Journal of Robotics.

Authors’ response: The paper has been checked by a native English speaker.

Reviewer 3 Report

Robotics

Title: Deep reinforcement learning for the control of robotic manipulation: a focussed mini-review

No.: robotics-1065618-peer-review-v1

This review paper offers research review on robotic manipulation control. Starting from deep learning and deep reinforcement learning, the current development of robotic manipulation control is provided. This paper also covers the review of algorithms for manipulation control.

The abstract is adequate. The structure of the paper is adequate. The paper is well-written, but the reviewer has 4 minor concerns.

(Concern 1)

The quality of Figure 2 is very bad, and the characters are too small. Please improve the quality of this figure.

(Concern 2)

In page 2, in line 49, the manuscript writes “The open-loop control does not not have external sensors or environment sensing capability, but heavily relies on highly structured environments that are very sensitively calibrated.”

“does not not have” needs to be modified.

(Concern 3)

In page 4, in line 118, the manuscript writes “It applies hundreds of thousands data-label pairs iteratively to train the parameters with loss computation and backprogation.”

“backprogation” needs to be modified.

(Concern 4)

Figure 9 is too dark to grasp. Please improve the quality of this figure.

(EOF)

Author Response

#3: Reviewer’s comments:

  1. The quality of Figure 2 is very bad, and the characters are too small. Please improve the quality of this figure.

 Authors’ response: As the reviewer suggested, Figure 2 has been improved for a higher quality.

  1. In page 2, in line 49, the manuscript writes “The open-loop control does not not have external sensors or environment sensing capability, but heavily relies on highly structured environments that are very sensitively calibrated.”

“does not not have” needs to be modified.

 Authors’ response: Thanks. We have modified into “does not have”

  1. In page 4, in line 118, the manuscript writes “It applies hundreds of thousands data-label pairs iteratively to train the parameters with loss computation and backprogation.”

“backprogation” needs to be modified.

Authors’ response: Thanks for pointing this out. We have corrected as “backpropagation”.

  1. Figure 9 is too dark to grasp. Please improve the quality of this figure.

Authors’ response: After discussion, we finally realized Figure 9 is not necessary for understanding our review paper. Also, to avoid copyright issues, we have now removed Figure 9 from the revised manuscript version.

Round 2

Reviewer 1 Report

the author corrects the manuscript according to the reviewer's comment.

Reviewer 2 Report

The authors accepted the comments, I recommend the paper to be published. Thanks.

Reviewer 3 Report

All required revisions have been made.